# *Orientia tsutsugamushi* Infection Stimulates Syk-Dependent Responses and Innate Cytosolic Defenses in Macrophages

**DOI:** 10.3390/pathogens12010053

**Published:** 2022-12-29

**Authors:** James Fisher, Casey Gonzales, Zachary Chroust, Yuejin Liang, Lynn Soong

**Affiliations:** 1Department of Microbiology and Immunology, University of Texas Medical Branch, Galveston, TX 77555-1070, USA; 2School of Medicine, University of Texas Medical Branch, Galveston, TX 77555-1070, USA; 3Department of Pathology, University of Texas Medical Branch, Galveston, TX 77555-1070, USA

**Keywords:** *Orientia*, obligate intracellular bacteria, innate immunity, pattern recognition receptor, Syk, C-type lectin, macrophage

## Abstract

*Orientia tsutsugamushi* is an obligately intracellular bacterium and an etiological agent of scrub typhus. Human studies and animal models of scrub typhus have shown robust type 1-skewed proinflammatory responses during severe infection. Macrophages (MΦ) play a critical role in initiating such responses, yet mechanisms of innate recognition for *O. tsutsugamushi* remain unclear. In this study, we investigated whether Syk-dependent C-type lectin receptors (CLRs) contribute to innate immune recognition and the generation of proinflammatory responses. To validate the role of CLRs in scrub typhus, we infected murine bone marrow-derived MΦ with *O. tsutsugamushi* in the presence of selective Syk inhibitors and analyzed a panel of CLRs and proinflammatory markers via qRT-PCR. We found that Mincle/*Clec4a* and *Clec5a* transcription was significantly abrogated upon Syk inhibition at 6 h of infection. The effect of Syk inhibition on Mincle protein expression was validated via Western blot. Syk-inhibited MΦ had diminished expression of type 1 cytokines/chemokines (*Il12p40*, *Tnf*, *Il27p28*, *Cxcl1*) during infection. Additionally, expression of innate immune cytosolic sensors (*Mx1* and *Oas1*-3) was highly induced in the brain of lethally infected mice. We established that *Mx1* and *Oas1* expression was reduced in Syk-inhibited MΦ, while *Oas2*, *Oas3*, and *MerTK* were not sensitive to Syk inhibition. This study reveals that Syk-dependent CLRs contribute to inflammatory responses against *O. tsutsugamushi*. It also provides the first evidence for Syk-dependent activation of intracellular defenses during infection, suggesting a role of pattern recognition receptor crosstalk in orchestrating macrophage-mediated responses to this poorly studied bacterium.

## 1. Introduction

Scrub typhus is a febrile illness caused by the obligately intracellular bacterium, *Orientia tsutsugamushi*, which is transmitted to humans via a bite of larval *Leptotrombidium* mites (commonly known as chiggers). Approximately 1 million scrub typhus cases occur per year in an Asia-Pacific region housing over one-third of the world’s population, termed the “tsutsugamushi triangle” [1]. However, recent reports have indicated the presence of scrub typhus in areas previously thought free of the disease, including South America and Africa [2,3]. The lung is a major target organ of infection and mild interstitial pneumonia predominates in self-limiting or appropriately treated cases [1]. However, if left untreated, disease may progress to severe lung damage and acute respiratory distress syndrome in up to 25% of cases, with a median fatality rate of 6% in untreated patients [1,4]. 

*O. tsutsugamushi* is a Gram-negative, lipopolysaccharide-negative coccobacillus, primarily infecting endothelial cells, monocytes, macrophages (MΦ), neutrophils, and dendritic cells [5]. Compared with other obligate intracellular bacteria such as *Rickettsia*, *Ehrlichia*, or *Anaplasma*, *O. tsutsugamushi* has unique yet poorly understood biology [6]. After the bacterium is internalized via endocytosis or phagocytosis, it rapidly (<4 h) escapes the endosome to freely inhabit the cytosol [5]. The bacteria utilize microtubules to traffic to the perinuclear region where replication occurs. *O. tsutsugamushi* replicates slowly, with peak rates occurring over 1–5 days post-infection and exits host cells via a budding mechanism [5,7]. A recent report has shown that *O. tsutsugamushi* actively inhibits NF-κΒ activation to evade host responses during its replication process [8]. However, very few reports have examined innate recognition of *O. tsutsugamushi* in different lifecycle phases or mechanisms of early cytokine responses to infection. 

Pattern-recognition receptor (PRR)-focused investigations for *O. tsutsugamushi* infection are scant. While Toll-like receptor 4 (TLR4) has no direct role, TLR2 was shown to sense *O. tsutsugamushi* in vitro [9]. However, in vivo experiments utilizing TLR2-deficient mice revealed a minor role in controlling bacterial growth and TLR2 was shown to promote susceptibility to lethal *O. tsutsugamushi* challenge via intraperitoneal inoculation [9]. A separate study demonstrated a role for MyD88 (a key adapter protein for several TLRs), as well as cytosolic nucleic acid sensors such as RIG-I (retinoic acid-inducible gene I), MAVS (mitochondrial antiviral signaling protein), and STING (stimulator of interferon response cGAMP interactor 1), in stimulating TNFα production in infected murine embryonic fibroblasts [10], but the biological function of these PRRs remain unclear. Likewise, the role of NLRs (nucleotide-binding site-leucine-rich repeat-like receptors) in generating inflammation has also been examined in vitro, but with conflicting results among studies [11,12,13]. The role of other cytosolic sensors of infection, including *Oas1-3*, remain poorly understood. Additionally, whether distinct PRRs are responsible for sensing extracellular versus intracellular *O. tsutsugamushi* is unknown. 

We recently reported a significant induction of C-type lectin receptors (CLRs), including Mincle/*Clec4e* and *Clec5a*, in the lungs and brain of C57BL/6 mice lethally infected with *O. tsutsugamushi,* as well as a significant upregulation of Mincle in infected bone marrow-derived MΦ and neutrophils [14]. Comparative studies of wild-type and Mincle^−/−^ MΦ further confirmed the role of Mincle in the regulation of proinflammatory chemokines/cytokines [14]. Mincle^−/−^ MΦ displayed reductions in neutrophil chemoattractant (*Cxcl1*) and macrophage chemoattractant (*Ccl2*) transcription at 4 h post-infection (hpi), after bacterial endosomal escape, along with nonsignificant reductions in *Tnf* and *Nos2*. At 24 hpi, when the bacterium has begun to replicate, there was a significant reduction in type 1-skewing cytokines/chemokines (*Il27*, *Cxcl9*, *Cxcl10*) in Mincle^−/−^ cells. Based on this in vivo and in vitro evidence, we proposed that Mincle upregulation contributed to the initiation and amplification of proinflammatory responses during *O. tsutsugamushi* infection. While this was the first report of CLR-mediated recognition of *O. tsutsugamushi*, the potential role of other individual CLRs or crosstalk of multiple CLRs or PRRs during infection remains unclear. 

The spleen tyrosine kinase (Syk) is an adaptor protein participating in the early events of CLR signal transduction [15]. Syk-coupled CLRs are expressed predominately in myeloid cells and can be found secreted or anchored to the plasma membrane [16,17]. This family of receptors recognizes endogenous and exogenous carbohydrate or glycolipid moieties [16]. CLRs interact with Syk via an immunoreceptor tyrosine-based inhibitory or immunoreceptor tyrosine-based activation motif in its own cytoplasmic tail, or through coupling with signaling partners (mainly FcγRs or DAP10/12) [16]. The phosphorylation of Syk recruits the Malt1/Card9/Bcl10 complex, which then activates NF-κΒ or AP-1 pathways, leading to context-specific inflammatory outcomes [16]. Given that Syk is utilized by many different CLRs, including those we have identified during *O. tsutsugamushi* infection (Mincle, Clec5a) [14], those identified during *Mycobacterium tuberculosis* infection (Mincle, Clec5a, Dectin-1, Dectin-2, Clec4d) [15,18], and others, the inhibition of Syk signaling provides a useful surrogate for examining CLR pathway activation [18,19]. Since gene-targeted Syk deletion is lethal in vivo, numerous small molecule inhibitors including Piceatannol (PIC), R406, and BAY 61-3606 (BAY), have been commercially developed to block Syk signaling with high selectivity and specificity. Each of these inhibitors has been utilized in vitro and in vivo to deduce redundancies in CLR signaling [20,21,22,23]. 

In this study, we tested whether CLR-mediated signaling contributes to proinflammatory responses during *O. tsutsugamushi* infection. Firstly, we utilized multiple Syk inhibitors to show that bone marrow derived-MΦ from C57BL/6 mice expressed Mincle and *Clec5a* in a Syk-dependent fashion. Then, utilizing the selective Syk inhibitor BAY, we demonstrated that transcription of type 1 responses in MΦ relied on intact Syk signaling. Furthermore, we identified differential regulation of cytosolic innate defenses including *Mx1* and *Oas1-3* during infection. To our knowledge, this is the first report revealing the regulation of Mincle and *Clec5a* expression and the impact of their signaling during *O. tsutsugamushi* infection.

## 2. Materials and Methods

### 2.1. Mouse Infection and Tissue Collection

Female C57BL/6J mice were purchased from Jackson Lab Inc. Animals were infected at 8-12 weeks of age, following Institutional Animal Care and Use Committee approved protocols (1902006) at the University of Texas Medical Branch (UTMB) in Galveston, TX, USA. All infections were performed in UTMB ABSL3 facilities in the Galveston National Laboratory, and subsequent tissue processing or analysis was performed in BSL3 or BSL2 facilities, respectively. Procedures were approved by the Institutional Biosafety Committee, in accordance with the Guidelines for Biosafety in Microbiological and Biomedical Laboratories. UTMB complies with the USDA Animal Welfare Act (Public Law 89-544), the Health Research Extension Act of 1985 (Public Law 99-158), the Public Health Service Policy on Humane Care and Use of Laboratory Animals, and NAS Guide for the Care and Use of Laboratory Animals (ISBN-13). UTMB is registered as a Research Facility under the Animal Welfare Act and has current assurance with the Office of Laboratory Animal Welfare, in compliance with NIH policy. The Karp strain of *O. tsutsugamushi* was utilized for all infections. Groups of 6 animals were intravenously infected with the same bacterial stock prepared from Vero cell infection, as described previously [24,25]. Mice were inoculated with a lethal dose of infection (~1.325 × 10^6^ viable bacteria, as determined via focus-forming assays) or PBS and monitored daily for weight loss and signs of disease. Brain samples were collected at 9–10 days post-infection (with mock-infected animals serving as controls) and inactivated for immediate or subsequent analyses. 

### 2.2. Infection of Mouse Bone Marrow-Derived Macrophages (MΦ)

Bone marrow cells were collected from the tibia and femur of mice and treated with red blood cell lysis buffer (Sigma Aldrich, Burlington, MA, USA). MΦ were generated by incubating bone marrow cells at 37 °C with 40 ng/mL M-CSF (Biolegend, San Diego, CA, USA) in complete RPMI 1640 medium (Gibco), as in our previous report [18]. Cell medium was replenished at day 3, and cells were collected at day 7. After collection and quantification, 5 × 10^5^ viable cells were seeded into 24-well plates or 2 × 10^6^ cells into 6-well plates and allowed to adhere overnight prior to infection. To determine Syk-dependent responses, three different chemical Syk inhibitors were tested. Each inhibitor was added to MΦs 30 min prior to infection at the following concentrations: 0.1, 0.5, and 1.0 μM BAY 61-3606 [21] (Sigma Aldrich); 0.5, 2.5, and 5.0 μM R406 [20,22] (Selleck Chemicals, Houston, TX, USA); and 5, 25, and 50 μM Piceatannol [23] (Selleck Chemicals). For IL-27-stimulated MΦ, recombinant mouse IL-27 (Biolegend) was added 30 min prior to infection at a final concentration of 100 ng/mL [26]. Bacteria were added at a multiplicity of infection (MOI) of 5 or 10 and centrifuged at 2000 RPM for 5 min to synchronize infection. The Karp strain of *O. tsutsugamushi* was utilized for all infections using the same bacterial stocks prepared from Vero cells, as described previously [24,27]. Experiments were performed in triplicate or quadruplicate. Infections were performed in BSL-3 facilities at the University of Texas Medical Branch, in Galveston, TX, USA. 

### 2.3. In Vitro Viability Staining

Bone marrow-derived MΦ (5 × 10^5^) were cultivated as described above, aliquoted into 15 mL conical tubes (Falcon), and allowed to rest for 1 h at 37 °C. Cells were then treated with 1.0 μM BAY 30 min prior to infection with 10 MOI *O. tsutsugamushi*. At 6 hpi, cells were washed once with PBS prior to incubation with live-dead stain (Invitrogen) for 15 min at 4 °C. Cells were then washed once with FACS buffer prior to fixation in 2% paraformaldehyde overnight at 4 °C. Data were collected by a BD LSRFortessa (Becton Dickinson, San Jose, CA, USA) and analyzed by using FlowJo software version 10.7.2 (Becton Dickinson).

### 2.4. Quantitative Reverse Transcription PCR (qRT-PCR) and RNA Sequencing

To determine host gene expression, MΦ cultures were collected in Trizol (Ambion, Austin, TX, USA) and incubated at 4 °C. overnight for inactivation. Total RNA was extracted via RNeasy mini kit (Qiagen, Germantown, MD, USA), and cDNA was synthesized utilizing an iScript cDNA kit (Bio-Rad Laboratories, Hercules, CA, USA). qRT-PCR assays were performed using iTaq SYBR Green Supermix (Bio-Rad) on a CFX96 Touch Real-Time PCR Detection System (Bio-Rad). The assay included: denaturing at 95 °C for 3 min followed with 40 cycles of 10 s at 95 °C and 30 s at 60 °C. To check the specificity of amplification, melt curve analysis was performed. Transcript abundance was calculated utilizing the 2^−ΔΔCT^ method and normalized to glyceraldehyde-4-phosphate dehydrogenase (GAPDH). Primers used in qRT-PCR analysis are listed in Appendix A. For transcriptomic analysis, 1 μg of total RNA from three infected and three mock C57BL/6 brain samples were submitted to LC Sciences, LLC (Houston, TX, USA) for RNA sequencing. 

### 2.5. Western Blot

Proteins from MΦ were extracted with RIPA lysis buffer (Cell Signaling Technology, Danvers, MA, USA) and quantified with a BCA Protein Assay kit (Thermo Fisher Scientific, Waltham, MA, USA). Samples were stored at −80 °C until processing. Thawed cell lysates were heated for 10 min at ~95 °C in Laemmli buffer (Bio-Rad) containing 2-β-mercaptoethanol, loaded into 4–20% SDS-PAGE gels (Bio-Rad) then transferred onto polyvinylidene difluoride membranes (Bio-Rad). After blocking, membranes were incubated with anti-Mincle (1:500, MBL International), anti-β-actin (1:2000, Cell Signalling Technology), and their corresponding anti-rabbit/goat secondary antibodies. Pierce ECL Western Blotting substrate (Thermo Fisher Scientific) was subsequently added to the membranes and light emission was captured using Amersham Imager 680 (GE Healthcare Lifesciences, Upssala, Sweden). Protein bands were quantified by using image analysis software (ImageJ). 

### 2.6. Statistical Analysis

Data were analyzed using GraphPad Prism software and presented as mean ± SEM. Differences between control and treatment groups were analyzed using one-way ANOVA with Dunnett’s multiple comparisons. Statistically significant values in comparison to mock samples are denoted as * *p* < 0.05, ** *p* < 0.01, *** *p* < 0.001, and **** *p* < 0.0001, respectively. Statistically significant values in comparison to *O. tsutsugamushi*-infected samples are denoted as # *p* < 0.05, ## *p* < 0.01, ### *p* < 0.001, and #### *p* < 0.0001, respectively.

## 3. Results

### 3.1. Syk Inhibition Reduces Expression of Mincle and Clec5a during O. tsutsugamushi Infection

Our previous report demonstrated a high degree of Mincle upregulation in *O. tsutsugamushi*-infected MΦ [14]. Since CLR activation has been shown to induce CLR transcription via a feed-forward mechanism [18], we sought to define the impact of Syk inhibition on CLRs during infection. To determine the dose-dependent effects of Syk inhibition, we added Syk inhibitors BAY (0.1, 0.5, 1.0 μM), PIC (5, 25, 50 μM), or R406 (0.5, 2.5, 5.0 μM) 30 min prior to infection with *O. tsutsugamushi* and assessed CLR gene transcription via qRT-PCR Western blot or at 6 h post-infection (hpi). First, we observed an infectious dose-dependent increase in Mincle protein detection at 6 hpi (Figure 1A), in congruence with our previous report [14]. Mincle expression was abrogated in the presence of 1 μM BAY (Figure 1A), with a ~45% reduction seen at 5 MOI and ~10% reduction at 10 MOI. We observed a statistically significant increase (*p* < 0.001) in Mincle transcription upon infection, which was abrogated in a dose-dependent manner in the presence of BAY and PIC, but not R406 (Figure 1B–D). Notably, Mincle transcription in infected cells was reduced by 80% with 1 μM BAY (Figure 1B) and by 85% with 50 μM PIC. (Figure 1C). We also observed a significant increase (*p* < 0.05) in *Clec5a* transcripts during infection, which was almost completely abrogated in the presence of 1.0 μM BAY (79% reduction) (Figure 1B) and 50 μM PIC (97% reduction) (Figure 1C), respectively. Clec12a displayed no significant changes, and Dectin-1 was significantly (*p* < 0.05) reduced during infection. Taken together, these results indicate that BAY is a highly potent Syk inhibitor in MΦ and that Mincle and *Clec5a* are the primary Syk-dependent CLRs induced upon *O. tsutsugamushi* infection.

### 3.2. Syk Is Critical for Neutrophil-Chemotactic Chemokines in MΦ

Syk signaling is important for the production of neutrophil and MΦ chemoattractants during infectious and non-infectious inflammatory processes [28]. We asked whether Syk signaling plays a role in chemoattractant production by infecting MΦ with *O. tsutsugamushi* (5 MOI) in the presence of the selective Syk inhibitor BAY. We observed a 27-fold increase (*p* < 0.001) in *Cxcl1* transcripts at 4 hpi and an 11-fold increase (*p* < 0.001) at 6 hpi (Figure 2A). Notably, transcripts of *Cxcl1* were reduced by 50% at 4 hpi and 60% at 6 hpi in the presence of 1 μM BAY. We then evaluated whether the reduction in *Cxcl1* levels due to Syk inhibition could be overcome by increasing the *O. tsutsugamushi* dose. Even after 10 MOI of the bacterium was added, we observed a statistically significant, dose-dependent decrease in *Cxcl1* levels at 6 hpi (~40% reduction at 0.5 μM; ~60% reduction at 1.0 μM BAY). Additionally, we observed a 19-fold increase (*p* < 0.001) in *Ccl2* transcripts at 4 hpi and a 30-fold increase (*p* < 0.001) at 6 hpi (Figure 2B) in cells exposed to 5 MOI of the bacterium. However, in contrast with *Cxcl1, Ccl2* transcript levels were not statistically significantly reduced in BAY-treated/infected cells. When 10 MOI of *O. tsutsugamushi* was utilized, we observed no significant reductions with 0.5 μM and a 40% reduction in *Ccl2* transcripts with 1 μM BAY. We then evaluated whether reductions in chemokine production could be attributed to altered cellular viability in the presence of BAY. We found that 87.1% of infected (10 MOI), BAY treated (1 μM) cells were viable at 6 hpi, compared with 98.3% of infected (10 MOI) untreated cells (Appendix A). Collectively, our results indicate that *Cxcl1* transcription is strongly Syk-dependent in *O. tsutsugamushi*-infected MΦs.

### 3.3. Type 1 Cytokine Production Is Reliant on Syk Signaling in MΦ

Monocytes and MΦs are significant sources of type 1 cytokines, including IL-12 and TNFα, during infectious processes [29,30]. Considering that animal models of scrub typhus display robust MΦ recruitment and type-1 responses in multiple organs [14,25,27], we evaluated whether Syk-dependent signaling may play a role in generating such responses in vitro. After infection with *O. tsutsugamushi* (5 MOI), we observed statistically significant increases in MΦ *Il12p40, Tnf,* and *Il27p28* transcript levels at 4 hpi, with peak expression for all cytokines occurring at 6 hpi (Figure 3). At 6 hpi, transcription of *Il12p40* was reduced by 80%, *Tnf* by 84%, and *Il27p28* by 94% in the presence of 1.0 μM BAY. MΦ infected with 10 MOI *O. tsutsugamushi* also displayed significantly high expression of *Il12p40, Tnf,* and *Il27p28.* While 0.1 μM BAY had small (*Il27p28*) or virtually no effect (*Il12p40* and *Tnf*), treatment with 1.0 μM BAY led to over 80% reduction in all tested cytokines in infection. Similarly, treatment with PIC (50 μM) virtually abolished *Il12p40* and *Tnf* expression (Appendix A). In contrast, R406 treatment (5 μM) significantly reduced *Tnf* transcripts by ~50%, with no major effect on *Il12p40* (Appendix A). These results indicate that type 1 cytokine production by infected MΦ is extensively the result of Syk-dependent signaling processes.

### 3.4. Expression of Innate Antiviral-like Genes Is Syk Dependent 

As an obligately intracellular pathogen, *O. tsutsugamushi* infection has been shown to elicit antiviral-like immune profiles [31]. Since *O. tsutsugamushi* replicates freely within the cytoplasm of host cells, intracellular innate defenses likely are activated during infection [10]. Yet, evidence for canonical RIG-I, cGAS, and STING activation has remained controversial [10]. In light of this, we tested whether cytosolic sensors *Oas1-3* and *Mx1* would be upregulated in response to infection. First, differential expressional analysis (RNA sequencing) of the infected C57BL/6 mouse brain at the terminal phase of disease (day 10) identified significant (adj. *p* < 0.05) upregulation of *Mx1* and *Oas1-3* compared with mock controls (Table 1). We then validated these findings using qRT-PCR, observing statistically significant increases in transcription of *Mx1*, *Oas1*, *Oas2*, and a near-significant increase in *Oas3* in brain tissue nearing terminal disease (day 9) (Figure 4A). We next examined whether these genes are expressed in infected MΦ, with MΦ survival gene MerTK serving as a control. At 6 hpi, *O. tsutsugamushi* infected (10 MOI) MΦ displayed statistically significant (*p* < 0.001) increases in *Mx1*, *Oas1*, and *Oas3* transcript levels, while no significant changes were observed for *Oas2* and MerTK (Figure 4B). We then tested whether gene transcription of *Oas1*-*3* and *Mx1* was dependent on Syk signaling since this previously has been demonstrated for viral infections [32]. Infected MΦ treated with 1 μM BAY displayed statistically significantly reduced levels of *Mx1* (79% reduction) and *Oas1* (49% reduction) at essentially all concentrations tested. Notably, BAY had no statistically significant impact on *Oas3, Oas2,* or MerTK transcripts. Together, these results demonstrate that select antiviral genes are upregulated in MΦ during *O. tsutsugamushi* infection and that *Mx1* and *Oas1* are Syk-dependent.

### 3.5. Innate Antiviral-like Responses Are Amplified by Recombinant IL-27

IL-27 has been shown to play an important role in stimulating the transcription of cytosolic innate immune sensors, such as *Mx1*, *Oas1,* and *Oas2*, during viral infection [26]. Considering that IL-27 is highly upregulated in *O. tsutsugamushi*-infected mouse lungs [14] and MΦ (Figure 3), we asked whether IL-27 may stimulate antiviral responses during infection. To address this question, we treated MΦ with 100 ng recombinant IL-27 prior to infection with *O. tsutsugamushi* (5 MOI) and assessed key markers at 6 hpi. We observed statistically significant increases in the transcription of *Mx1*, *Oas1*, *Oas3,* and *Oas2* during infection (Figure 5), consistent with Figure 4. Interestingly, we found that IL-27 stimulation prior to infection led to a significant increase in *Mx1* expression, which was greater than infection alone. While *Oas1* and *Oas3* displayed a similar trend, the result did not meet statistical significance. Notably, IL-27 did not alter the expression of any markers tested in uninfected cells, with the exception of a marginal increase in MerTK. Thus, IL-27 enhances the transcription of *Mx1* during *O. tsutsugamushi* infection.

## 4. Discussion

Unlike other Gram-negative bacteria, *O. tsutsugamushi* lacks classical PAMPs such as lipopolysaccharide, canonical peptidoglycan, or flagellin; consequently, mechanisms of innate immune recognition have remained puzzling. This study was aimed at identifying redundancies in PRR signaling by targeting the CLR adaptor protein Syk. Using primary murine MΦ, we revealed that Mincle and *Clec5a* are highly transcribed and Syk-dependent during infection. We also revealed that the production of type 1 cytokines, including *Il12p40*, *Tnf,* and *Il27p28*, relies on intact Syk signaling. Finally, we demonstrated that infection induces transcription of cytosolic innate immune sensors *Mx1*, *Oas1*, and *Oas3* in vivo and in vitro. This study is important in several aspects, suggesting that MΦ proinflammatory responses to infection are largely generated via Syk-dependent processes. 

Firstly, we observed that *O. tsutsugamushi* infection highly upregulated Mincle in MΦ at 6 hpi, along with *Clec5a*, in a Syk-dependent manner (Figure 1). In contrast, other tested CLRs were reduced (Dectin-1) or displayed no transcriptional changes (*Clec12a*). These findings support our previous report showing Mincle as the most highly upregulated CLR in vivo and in vitro, followed by *Clec5a* [14]. Activation of Mincle, *Clec5a*, and Dectin-1 (among others) initiates downstream signaling via the adaptor protein Syk [18,33] and reports have shown this process may lead to increased CLR transcription via a feed-forward mechanism [18]. In agreement with this notion, we observed dose-dependent decreases in Mincle and *Clec5a* transcription when Syk was inhibited with BAY and PIC during infection (Figure 1). However, Mincle and *Clec5a* transcript levels were not completely reduced, even at the highest concentrations of inhibitors used. This indicated transcriptional regulation of these CLRs was partially reliant on Syk signaling and suggested Mincle and *Clec5a* may be regulated by other factors, including cytokines or DAMPs. The importance of TNF in CLR regulation was recently established, as recombinant TNF stimulation alone was shown to induce Mincle transcription and expression in MΦ [14,34]. Accordingly, we observed that *Tnf* transcription was not completely abolished, even at the highest concentration of inhibitors used (Figure 3 and Appendix A). Thus, it is plausible that *Tnf* expression via Syk-independent mechanisms stimulates the transcription of Mincle and *Clec5a*. 

We were surprised that R406 had minimal effect on the transcription of the tested CLRs. Whether this is due to differences in the selectivity of the inhibitors used or other biologic processes, such as drug metabolism, remains to be defined. Though commonly used in CLR research, Syk inhibitors display differential selectivity and potencies. PIC, a naturally occurring stilbene, was one of the first Syk inhibitors identified [35] and arguably has been the most widely used inhibitor in CLR studies [36,37,38]. However, PIC also weakly inhibits other tyrosine kinases such as Lyn, cAK, and PKC (Manufacturer Note, Sigma Aldrich). R406 is a highly selective Syk inhibitor developed for clinical use, but may also inhibit Flt3 (Manufacturer Note, Selleck Chemical). BAY is regarded as one of the most selective and potent inhibitors, with no established off-target inhibition (Manufacturer Note, Selleck Chemical). Our parallel comparison of BAY, PIC, and R406 largely confirm this notion. We found BAY treatment significantly reduced Mincle and *Clec5a* expression during infection at much lower concentrations than PIC or R406 (Figure 1). Notably, the decrease in expression of Mincle and *Clec5a* was unlikely due to reductions in viability between the infected comparison groups (Appendix A). This allowed us to focus on utilizing BAY for the remainder of our in vitro studies. Future efforts examining the impact of in vivo Syk inhibition with BAY could prove highly useful for examining the role of multiple CLRs in scrub typhus pathogenesis.

Secondly, this study presents the first evidence for Syk-dependent MΦ responses during *O. tsutsugamushi* infection. MΦs play key roles in infection with *O. tsutsugamushi* and other closely related *Rickettsia* species, both as perpetrators of tissue damage and as target cells for bacterial replication [39,40,41,42]. We demonstrated here that MΦ *Cxcl1* transcription, a key chemokine for neutrophil recruitment, was highly sensitive to Syk inhibition (Figure 2). In contrast, *Ccl2* (a key chemokine for macrophage recruitment) transcription was less sensitive and only reduced by ~40% at the maximal concentration of BAY. Thus, MΦ-mediated neutrophil recruitment may heavily rely on Syk signaling. This finding is important as extensive leukocyte recruitment is thought to play a prominent role in tissue damage during scrub typhus [24,25,27]. Examining the effect of Syk signaling in murine models of scrub typhus via selective Syk inhibition or SYK^fl/fl^ LysM-Cre systems is an important emphasis for future studies. Additionally, evaluating whether Syk-inhibited MΦs permit or restrict bacterial growth will shed further insight into the role of multiple CLRs controlling the infection.

We were surprised to find that MΦ transcription of major type 1 cytokines, including *Tnf*, *Il12p40*, and *Il27p28* were all reduced by more than 80% when Syk was inhibited during infection (Figure 3). This is notable since our previous report utilizing single CLR-deficient (e.g., Mincle^−/−^) MΦs displayed nominal reductions in *Tnf* and *Il27* transcripts early in infection (4 hpi) [14]. While our study was limited to transcriptional analysis, our data aligns closely with a report examining type 1 cytokine production generated via J774.1 cells (a MΦ-like cell line derived from the BALB/c mouse background). These authors found inhibition of ERK, JNK, and p38 abrogated TNFα production by ~80% during *O. tsutsugamushi* infection [43]. However, ERK, JNK, and p38 activation occurs multiple steps downstream of Syk activation and via multiple PRR signaling pathways. Our study permitted the evaluation of CLR-specific signaling during infection and implied a potentially important role of multiple CLRs in generating inflammation. Considering that overzealous proinflammatory/type 1 responses are thought to contribute to indiscriminate tissue damage in scrub typhus, our study presents a strong premise for future investigation into CLR engagement during disease. 

Thirdly, we revealed the upregulation of intracellular nucleic acid sensors *Mx1* and *Oas1-3* in infected mouse brains and MΦs (Table 1, Figure 4 and Figure 5). Our findings corroborate a report that revealed increased expression of *Oas1* and *Mx1* in infected human peripheral blood mononuclear cells [44]. Additionally, dual-RNA sequencing has uncovered a high degree of *Mx1* and *Oas1* gene expression in *O. tsutsugamushi*-infected human umbilical vein endothelial cells [31] and a genome-wide association study of scrub typhus patients recently implicated *Oas1* in susceptibility to disease [45]. OAS family proteins sense double-stranded RNA in the cytoplasm and have been most studied in the context of viral infection [46]. To our knowledge, only a single report has examined the contribution of OAS proteins during bacterial infection, which revealed an important role for these proteins in restricting intracellular *Mycobacterium tuberculosis* growth [47]. Since *Mx1*, *Oas1,* and *Oas2* were upregulated in cells that *O. tsutsugamushi* preferentially targets for replication (endothelial cells and MΦ), future studies utilizing knockdown or genetic deletions of these molecules are needed to delineate their role during infection. 

Additionally, transcription of *Mx1* and *Oas* members has recently been shown to be regulated by multiple avenues, including both Syk-dependent [32] and IL-27-dependent [26] processes. We were intrigued to find that *Mx1* and *Oas1* transcription was reduced when Syk was inhibited, while *Oas3* was not significantly impacted (Figure 4). This indicates potentially divergent signaling pathways for the transcriptional regulation of these markers. Additionally, IL-27 stimulation prior to infection led to significantly increased *Mx1* transcription in our study, but it did not significantly increase *Oas1-3* transcripts (Figure 5). This finding contrasts with a previous report, which revealed IL-27 alone in dramatically increasing *Oas1* expression in the context of Zika virus infection [26]. Whether this finding represents differences in cell types used or biological differences between target host cells of these pathogens remain unknown. 

Based on our findings presented herein, we propose a hypothetical model for the contribution of Syk-dependent signaling in MΦ responses to *O. tsutsugamushi* (Figure 6). MΦs can sense the bacterium via Syk-dependent receptors, including Mincle, Clec5a, and potentially others, at the initial stages of infection. MΦs increase Mincle expression via feed-forward mechanisms, TNFα-mediated mechanisms, or stimulation with DAMPs as the infection progresses. Syk-dependent transcription of intracellular defense-related genes (*Oas1*, *Mx1*), neutrophil chemoattractant (*Cxcl1*), and Th1-skewing cytokines (*Tnf*, *Il27*, *Il12*) ensues. Excessive Syk activation via *O. tsutsugamushi*, host DAMPs, or TNFα in the microenvironment, can result in sustained inflammatory responses. These innate responses may ultimately contribute to Th1/CD8-skewed responses and acute tissue damage, as we observed in the lungs and other tested organs [27,48,49]. Future investigation with targeted blockage of Syk, Mincle, or Clec5a in the host will help define the immunologic impact of CLR cooperation on scrub typhus. Such studies may help understand immune responses against other obligately intracellular pathogens, including *Rickettsia*, *Ehrlichia*, and *Chlamydia*, for which virtually no information is available regarding the role of CLRs in pathogen recognition.

In summary, this study provided new insight into the activation of CLR pathways for innate immune recognition of *O. tsutsugamushi* and identified the expression of intracellular sensors implicated in cytosolic defense. Through MΦ infection in the presence of Syk inhibitors, we provided the first evidence for Syk-dependent, type 1- and antiviral-like responses. While our studies were limited to transcriptional and predominantly in vitro data, future studies will evaluate the impact of Syk signaling in vivo. To date, PRR redundancies represent a virtually unexplored aspect of *O. tsutsugamushi* immunology. Considering the relatively low degree of PRR activation observed in numerous reports, it is conceivable that multiple receptors are engaged during infection. The parallel engagement of numerous PRRs may contribute to or drive overzealous immune responses. We speculate that CLR crosstalk and redundancies may contribute to malicious immune responses and progressive tissue damage in scrub typhus. A better understanding of how CLR pathway activation occurs and which CLRs are responsible for immune regulation in vivo may aid the design of treatments or vaccines for severe scrub typhus. 

## Figures and Tables

**Figure 1 pathogens-12-00053-f001:**
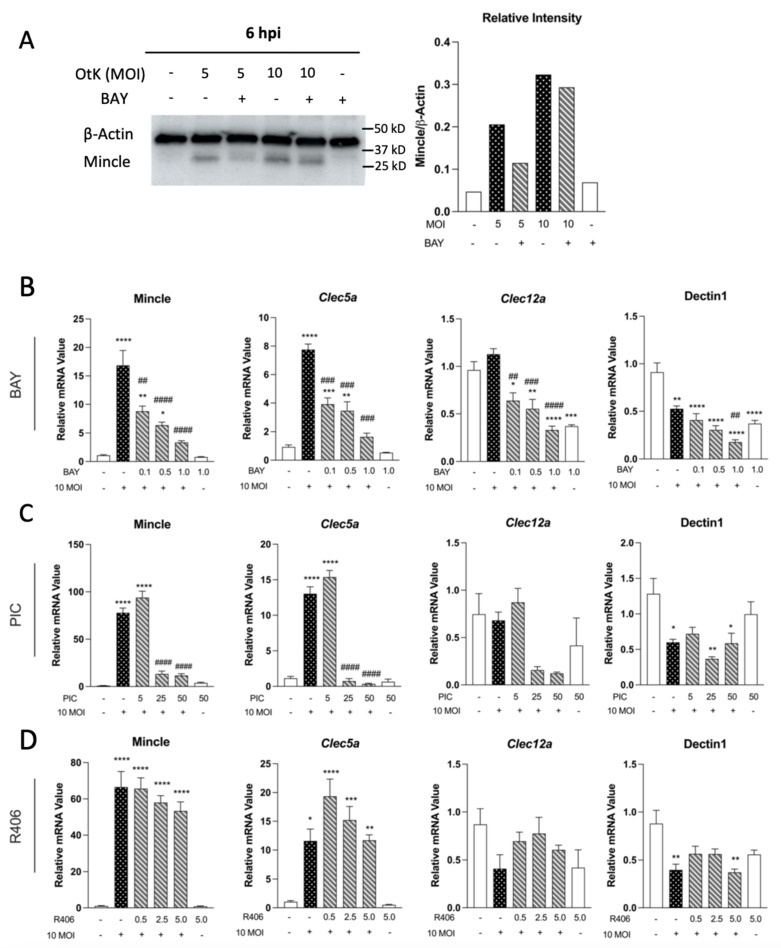
MΦ CLR expression is dependent on Syk signaling during *O. tsutsugamushi* infection. Bone marrow-derived MΦ were treated with different concentrations of Syk-inhibitors for 30 min prior to infection with *O. tsutsugamushi* (10 MOI). (**A**) WB and densitometry analysis of MΦ infected at 5 or 10 MOI in the presence of 1 µM BAY (**B**) Effect of BAY (0.1, 0.5, 1.0 µM) (**C**) Effect of PIC (5, 25, and 50 µM) (**D**) Effect of R406 (0.5, 2.5, 5.0 µM) on CLRs at 6 hpi. qRT-PCR analyses of target genes relative to GAPDH are shown as mean ± SEM. One-way ANOVA with Dunnett’s multiple comparison test was performed across all treatment groups. For groups in comparison to mock controls: *, *p* < 0.05; **, *p* < 0.01; ***, *p* < 0.001; ****, *p* < 0.0001. For infected-and-treated groups in comparison to infected-but-untreated samples: ##, *p* < 0.01; ###, *p* < 0.001, ####, *p* < 0.0001.

**Figure 2 pathogens-12-00053-f002:**
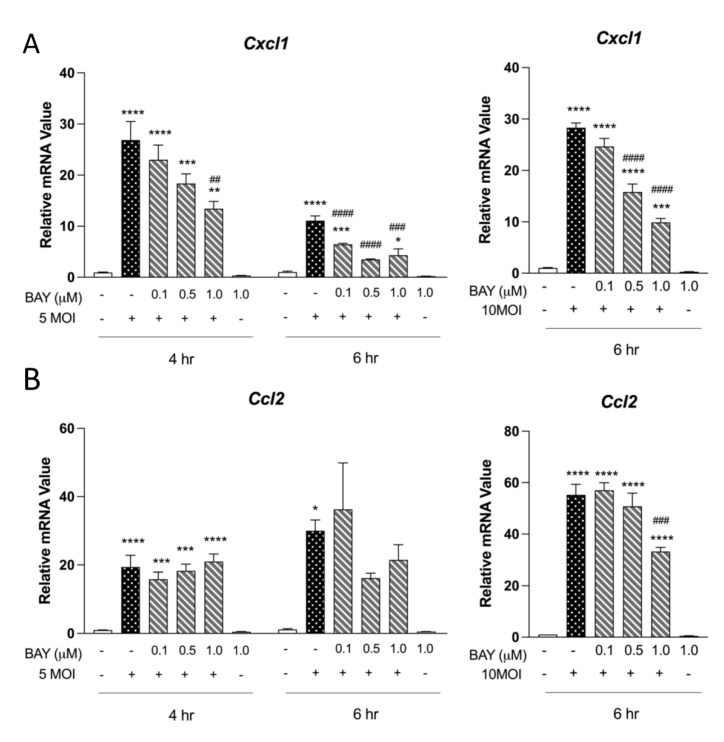
MΦ Cxcl1 expression relies on intact Syk signaling during *O. tsutsugamushi* infection. Bone marrow-derived MΦ were treated with Syk-inhibitor BAY (0.1, 0.5, 1.0 µM) for 30 min prior to infection with *O. tsutsugamushi* (5 or 10 MOI). qRT-PCR analyses of (**A**) *Cxcl1* and (**B**) *Ccl2* relative to GAPDH are shown as mean ± SEM. One-way ANOVA with Dunnett’s multiple comparison test was performed across all treatment groups. For groups in comparison to mock controls: *, *p* < 0.05; **, *p* < 0.01; ***, *p* < 0.001; ****, *p* < 0.0001. For infected-and-treated groups in comparison to infected-but-untreated samples: ##, *p* < 0.01; ###, *p* < 0.001; ####, *p* < 0.0001.

**Figure 3 pathogens-12-00053-f003:**
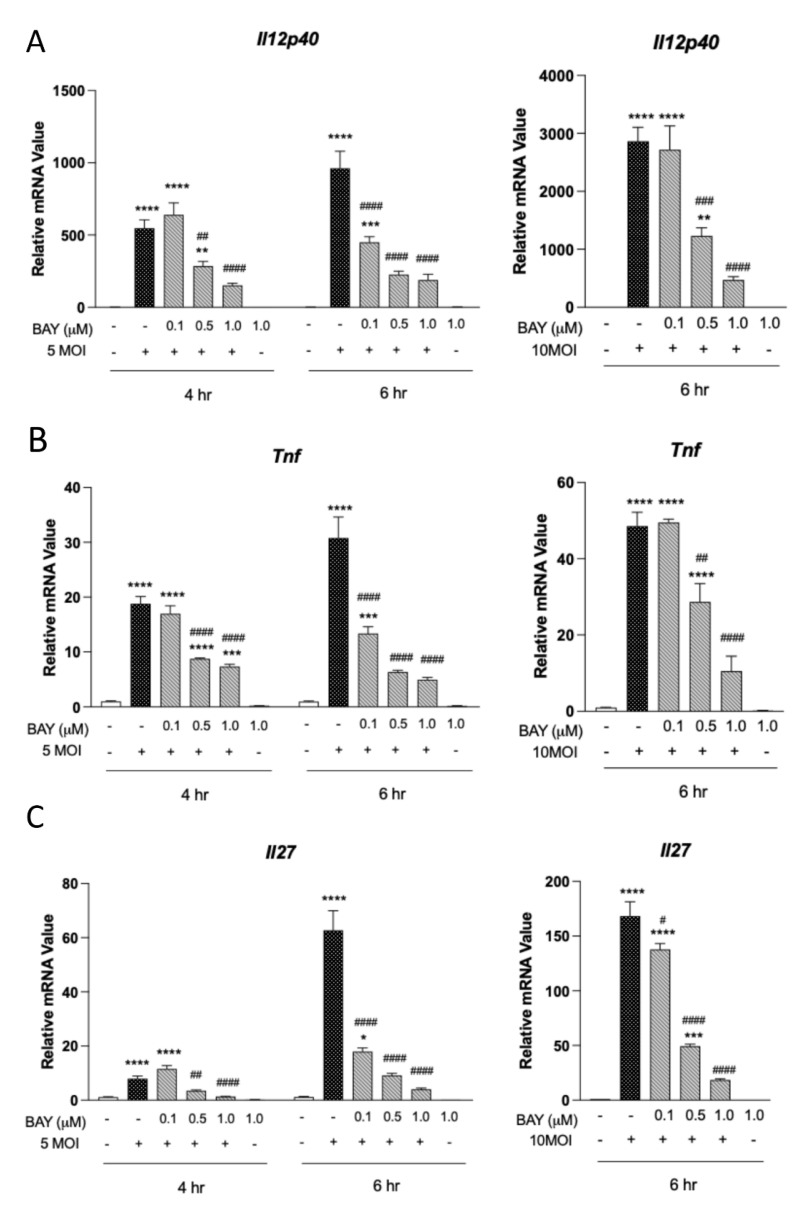
Syk signaling drives type 1 responses in *O. tsutsugamushi*-infected MΦ. Bone marrow-derived MΦ were treated with Syk-inhibitor BAY 61-3606 (0.1, 0.5, 1.0 µM) for 30 min prior to infection with *O. tsutsugamushi* (5 or 10 MOI). qRT-PCR analyses of (**A**) *Il12p40*, (**B**) *Tnf*, and (**C**) *Il27p28* relative to GAPDH are shown as mean ± SEM. One-way ANOVA with Dunnett’s multiple comparison test was performed across all treatment groups. For groups in comparison to mock controls: *, *p* < 0.05; **, *p* < 0.01; ***, *p* < 0.001; ****, *p* < 0.0001. For infected-and-treated groups in comparison to infected-but-untreated samples: #, *p* < 0.05; ##, *p* < 0.01; ###, *p* < 0.001, ####, *p* < 0.0001.

**Figure 4 pathogens-12-00053-f004:**
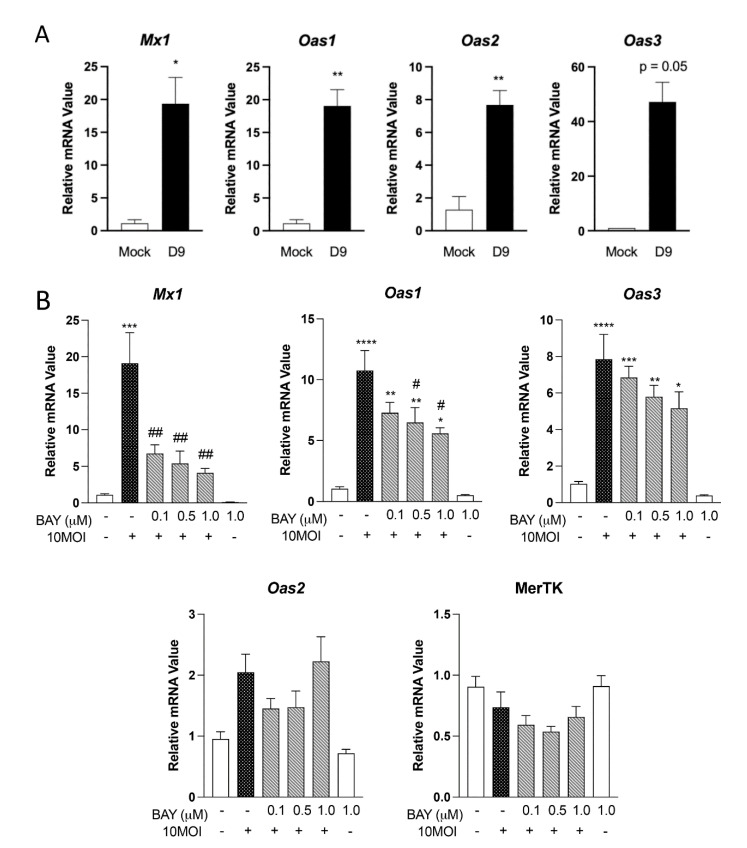
Antiviral-like responses are upregulated in vivo and Syk-dependent in vitro. (**A**) qRT-PCR detection of indicated antiviral genes from whole brain tissue homogenates of lethally infected C57BL/6 mice. (**B**) Bone marrow-derived MΦ were treated with Syk-inhibitor BAY (0.1, 0.5, 1.0 µM) for 30 min prior to infection with *O. tsutsugamushi* at 10 MOI. qRT-PCR analyses of select antiviral markers are shown at 6 hr post infection. One-way ANOVA with Dunnett’s multiple comparison test was performed across all treatment groups. For groups in comparison to mock controls: *, *p* < 0.05; **, *p* < 0.01; ***, *p* < 0.001; ****, *p* < 0.0001. For infected-and-treated groups in comparison to infected-but-untreated samples: #, *p* < 0.05 ##, *p* < 0.01.

**Figure 5 pathogens-12-00053-f005:**
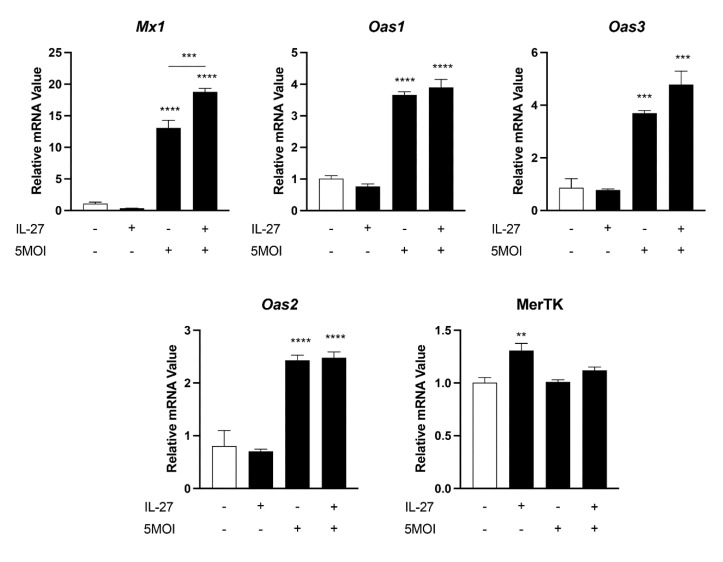
IL-27 amplifies Mx1 expression in response to *O. tsutsugamushi*. Bone marrow-derived MΦ were stimulated with recombinant IL-27 (100 ng) for 30 min prior to infection with *O. tsutsugamushi* at 5 MOI. qRT-PCR analyses of select antiviral markers are shown at 6 hr post-infection. One-way ANOVA with Dunnett’s multiple comparison test was performed across all treatment groups. **, *p* < 0.01; ***, *p* < 0.001; ****, *p* < 0.0001.

**Figure 6 pathogens-12-00053-f006:**
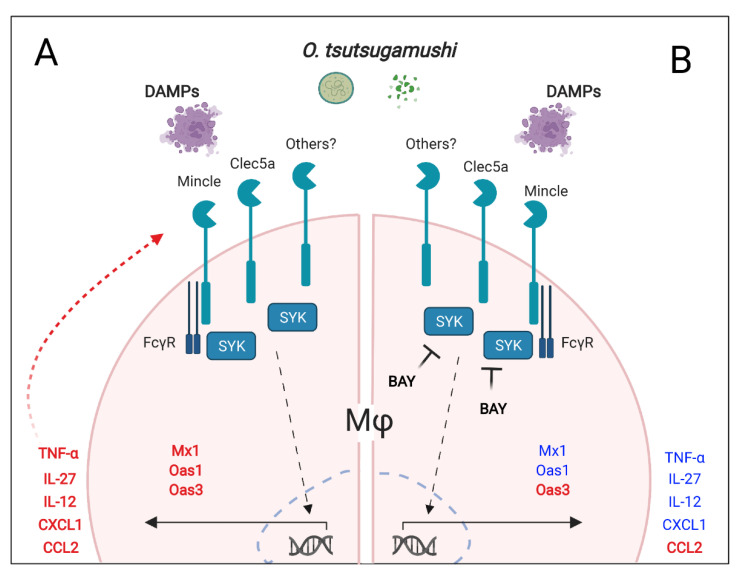
A schematical model for the role of Syk-dependent CLR responses to *O. tsutsugamushi* infection. *O. tsutsugamushi* bacterial components, damaged host molecules (DAMPs), and TNFα can stimulate Mincle and Clec5a, leading to Syk activation. (**A**) Collective signaling results in an increased expression of type 1 cytokines (TNFα, IL-27, IL-12), chemokines (CXCL1, CCL2), and intracellular defenses (Mx1, Oas1, Oas3). (**B**) Syk-dependent responses (in blue) include the production of key type 1 cytokines (TNFα, IL-27, IL-12), neutrophil chemoattractant CXCL1, and intracellular defenses (Mx1, Oas1). Collectively, these events attempt to control bacterial growth via promoting type 1 or antiviral-like responses, but excessive activation may exacerbate cellular damage.

**Table 1 pathogens-12-00053-t001:** Expression of select nucleic acid sensors in the brain (D10 vs. mock) determined via RNA sequencing.

Gene	Fold Change
Oas3	35.99 *
Mx1	33.03 *
Oas1b	14.16 *
Oas2	7.90 *

*, adj. *p* < 0.05.

## Data Availability

Not applicable.

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
