# Peer review of "Orientia tsutsugamushi Infection Stimulates Syk-Dependent Responses and Innate Cytosolic Defenses in Macrophages"

_pathogens, 2022, doi:10.3390/pathogens12010053_

Round 1
Reviewer 1 Report
Please see attached file.

Author Response
We appreciate the reviewers for asking thoughtful questions and providing helpful suggestions. Please see our reply attached.

Reviewer 2 Report
Orientia tsutsugamushi infection stimulates Syk-dependent responses and innate cytosolic defenses in macrophages
The authors previous described a mechanism by which Orientia tsutsugamushi modulated the C-type lectin receptor Mincle and type 1-proinflammatory immune responses in mice and bone marrow-derived macrophages (PMID: 34320039). This manuscript builds on the previous findings and attempts to evaluate the role Syk-mediated signaling via CLRs in regulating immune recognition and generation of proinflammatory responses in macrophages.
· Using O. tsutsugamushi-infected murine bone marrow-derived in the presence or absence of selective Syk inhibitors the authors analyzed a panel of CLRs and proinflammatory markers via qRT-PCR and showed that Mincle (Clec4a) and Clec5a transcripts were abrogated upon Syk inhibition.
· Syk inhibition (BAY) modulated Mincle protein, while Syk-inhibition (BAY) diminished in a concentration-dependent manner the mRNA expression values of type 1 cytokines/chemokines (Il12p40, Tnf, Il27p28, Cxcl1).
· Also, using mouse brain homogenates the authors showed that O. tsutsugamushi infection resulted in an induction of mRNA expression of innate immune cytosolic sensors (Mx1 and Oas1-3). In addition, Syk-inhibition (BAY) in Mf diminished the expression of Mx1 and Oas1 expression, while Oas2, Oas3, and MerTK were not affected.
Overall, the work by Fisher et al., provide interesting data., suggesting a Syk-dependent CLRs contribution to inflammatory responses upon O. tsutsugamushi in macrophages, however the manuscript could benefit from additional data sets to improve enthusiasm and scientific rigor. Specifically, most presented data are highly correlative and the authors do not present sufficient functional data to demonstrate that the presented data actually have a biological role in modulating O. tsutsugamushi infectivity and pathogenicity.
Major comments:
No data were provided to address whether Syk treatment is affecting O. tsutsugamushi internalization and replication within macrophages and animal tissues.
The authors provided conclusions exclusively rely on experiments that utilize pharmacological inhibitors to block Syk-dependent signaling events. Although the authors provide some supporting rational that the inhibitors are in theory can be considered to specifically inhibit Syk activity, off target effects can not be ruled out. So experiments using mouse models that are deficient in Syk expression should be considered. Given the authors focus on macrophages conditional SYKfl/fl LysM-cre mice are commercially available and therefore should be considered to provide findings that further support the authors conclusions.
Figure 1:
A) The authors assessed the Mincle protein expression via western blot but did not provide any quantification of the observed difference in Mincle expression in the presence or absence of BAY. Corresponding molecular weights of the shown molecules should also be provided. No O. tsutsugamushi specific protein was used as loading control to demonstrate the successful infection of the macrophages with O. tsutsugamushi.
B) The authors show qRT-PCR data of CLRs upon the pharmacological inhibition of Syk during O. tsutsugamushi infection. However, how do the authors account for the potential differences in bacterial loads upon host cell infection, which consequently could account for the observed difference in mRNA expression of the target genes? No description on how the authors assess bacterial loads during these experiments was provided within the manuscript (not in the material and method section or anywhere else). Similar concerns apply for most other Figures (Figures 2-5).
Figure 2:
The authors try to claim that Cxcl1 expression relies on intact Syk signaling during O. tsutsugamushi infection. However, the authors only show a correlation between inhibition of Syk via pharmacological inhibition (BAY) and a decrease in Cxcl1 not Ccl2 mRNA expression. No signaling events were assessed. Foremost, the authors never demonstrated that Mincle, FcyR and Syk form a complex during O. tsutsugamushi infection, and that this association is diminished upon Syk-inhibition. Also, the authors should evaluate whether the observed decrease in mRNA level results in a decrease in the production/secretion of the selected chemokines.
Figure 3: As mentioned for data in Figure 2, the authors should assess whether the observed decrease in mRNA level of the selected cytokines results in a decrease in the production/secretion of the chemokines (e.g., by flow cytometry).
Figures 4/5: Observed changes in mRNA levels of anti-viral genes (in macrophages and/or mouse tissue homogenates) should also be assessed at the protein level. Overall, the findings in both figures seem interesting but no functional data were provided to link the observed findings to O. tsutsugamushi pathogenicity. For instance: how is an IL-27-mediated amplification of Mx1 beneficial to O. tsutsugamushi infection of the host
or in more general terms how is Mx1 expression effecting O. tsutsugamushi infection…
Minor concerns:
Figure 1-3 legends contain spelling mistakes
Figure 6: macrophage symbol is incorrect
Tnf should be changed to Tnfa
Author Response
We appreciate reviewer 2 for asking thoughtful questions and providing helpful suggestions. These great suggestions are also the future direction for our studies.
Please see our reply attached.

Round 2
Reviewer 2 Report
Accept in present form